# Clinical Use of Lansoprazole and the Risk of Osteoporosis: A Nationwide Cohort Study

**DOI:** 10.3390/ijerph192215359

**Published:** 2022-11-21

**Authors:** Ming-Hsuan Chung, Yong-Chen Chen, Wen-Tung Wu, Ming-Hsun Lin, Yun-Ju Yang, Dueng-Yuan Hueng, Tsung-Kun Lin, Yu-Ching Chou, Chien-An Sun

**Affiliations:** 1Department of Neurological Surgery, Tri-Service General Hospital, National Defense Medical Center, Taipei 114, Taiwan; 2Data Science Center, College of Medicine, Fu-Jen Catholic University, New Taipei City 242, Taiwan; 3Department of Medicine, College of Medicine, Fu-Jen Catholic University, New Taipei City 242, Taiwan; 4Department of Pharmacy, Tri-Service General Hospital, National Defense Medical Center, Taipei 114, Taiwan; 5School of Pharmacy, National Defense Medical Center, Taipei 114, Taiwan; 6Division of Endocrinology and Metabolism, Department of Internal Medicine, Tri-Service General Hospital, National Defense Medical Center, Taipei 114, Taiwan; 7School of Public Health, National Defense Medical Center, Taipei 114, Taiwan; 8Department of Public Health, College of Medicine, Fu-Jen Catholic University, New Taipei City 242, Taiwan

**Keywords:** cohort study, lansoprazole, osteoporosis, proton pump inhibitors

## Abstract

Background: Proton pump inhibitor (PPI) lansoprazole acts as a liver X receptor agonist, which plays a crucial role in the crosstalk of osteoblasts and osteoclasts in vitro and during bone turnover in vivo. However, epidemiological studies on the association between the use of lansoprazole and osteoporosis risk are limited. We aimed to determine the risk of developing osteoporosis in patients with lansoprazole use. Methods: A retrospective cohort study was conducted using the National Health Insurance Research Database of Taiwan dated from 2000 to 2013. The study includes 655 patients with lansoprazole use (the exposed cohort) and 2620 patients with other PPI use (the comparison cohort). The main outcome was the primary diagnosis of osteoporosis. The hazard ratios (HRs) and 95% confidence intervals (CIs) were used to assess the association between the use of lansoprazole and risk of osteoporosis. Results: Patients receiving lansoprazole treatment had a reduced risk of osteoporosis as compared with those undergoing other PPI therapy (adjusted HR, 0.56; 95% CI, 0.46–0.68). Moreover, this inverse association is evident in both sexes and in various age groups. Conclusions: This population-based cohort study demonstrated that lansoprazole use was associated with a reduced risk of osteoporosis. The clinical implications of the present study need further investigations.

## 1. Introduction

Osteoporosis, characterized by the micro-architectural deterioration of bone tissue and loss of bone mass, increases the risk of bone fracture and can lead to substantial long-term disability and decreased quality of life [1]. Many risk factors of osteoporosis have been identified, including ethnicity, sex, body mass index, physical activity, and smoking [2]. There are also a number of therapeutic agents, such as corticosteroids, selective serotonin receptor inhibitors, thiazolidinedione, anticonvulsants, calcineurin inhibitors, and anticoagulants, that are associated with drug-induced osteoporosis [3]. Recently, there has been concern about the effect of proton pump inhibitors (PPIs) on the risk of osteoporosis. PPIs are extremely effective in reducing acid secretion by irreversibly blocking the hydrogen/potassium adenosine triphosphatase enzyme system (the H^+^/K^+^ ATPase or the gastric proton pump) of gastric parietal cells [4]. However, stomach acids are necessary to absorb calcium, proteins, vitamin B12, drugs, and other nutrients. Therefore, in conditions of prolonged hypochlorhydria, their absorption can result in impairment. PPIs seem to be associated with a risk of osteoporosis, with a primary potential mechanism involving the physiological effects of chronic acid suppression on calcium, magnesium, and parathyroid hormone metabolism [5]. In addition, PPI may cause dose-dependent inhibitory effects on osteoclastic and osteoblastic human cells, leading to a kind of low bone turnover syndrome. Therefore, PPI-related bone fragility might be determined by the impairment of the repair mechanisms for bone microfractures [6,7]. In particular, Cheng et al. identified that lansoprazole, a commonly used PPI, could inhibit the P-type ATPases SERCA and Ca^2+^-ATPase, leading to an increase in Ca^2+^ in osteoblasts and inducing endoplasmic reticulum stress and apoptosis; furthermore, blocking the increase in intracellular calcium had a significant protective effect against osteoblast apoptosis [8]. Data on the effect of PPIs on osteoporotic fracture risk have, however, been divergent. There are epidemiologic analyses demonstrating a significant association between PPI use and osteoporotic fractures [9,10,11]. A meta-analysis from 2019 with 24 observational studies and 2,103,800 participants concluded that there was a modestly increased risk of hip fractures in those treated with PPIs [12]. However, not all studies evaluating PPIs and osteoporosis have demonstrated this positive association [13,14]. Because PPIs are the most commonly used gastric-acid-suppressing medications, any effect of PPI use on the development of osteoporosis would have significant public health implications.

Among PPIs, lansoprazole has anti-glycemic effects in patients with type 2 diabetes mellitus [15,16]. In addition, lansoprazole has been reported to induce adipocyte differentiation and increase insulin sensitivity [17]. Recently, a cross-sectional study of nationally representative samples of Korean men indicated that insulin resistance and fasting hyperinsulinemia were inversely associated with bone mass in men [18]. Apparently, the use of lansoprazole may be related to bone mass through the insulin pathway. In addition, it has been reported that the liver X receptor (LXR) plays a crucial role in the crosstalk of osteoblasts and osteoclasts in vitro and during bone turnover in vivo [19]. Consequently, alterations in LXR have been observed to play a crucial role in physiological and pathological bone turnover [20]. An LXR agonist would have therapeutic potential in treating bone diseases. It has been noted that lansoprazole acts as an LXR agonist [21]. Given these study results, we sought to evaluate the association between the clinical use of lansoprazole and the risk of osteoporosis using data from the National Health Insurance Research Database (NHIRD) in Taiwan.

## 2. Materials and Methods

### 2.1. Data Source

Taiwan launched a national single-payer compulsory enrollment health insurance program in 1995. The NHIRD covers 99% of the Taiwanese population, includes all claims data of the Taiwanese and makes these data available to scientists in Taiwan for research purposes. The NHIRD contains comprehensive healthcare information, including demographic data of insured individuals, data of clinical visits, diagnostic codes, and prescription details. The diagnostic codes used in the NHIRD were in the format of the International Classification of Diseases, Ninth revision, Clinical Modification (ICD-9-CM). NHIRD has been used for high-quality epidemiological studies and has shown a good validity [22,23,24]. The data of this study was obtained from the Longitudinal Health Insurance Database 2000 (LHID 2000), a subset of NHIRD. The LHID 2000 dataset contains historical ambulatory and inpatient care data for 1 million randomly sampled beneficiaries enrolled in the National Health Insurance system in 2000. The LHID 2000 database allows researchers to approach the history of using medical services in these patients. There were no significant differences in the distributions of age, sex, and healthcare costs between the individuals in LHID and NHIRD [22]. Since the dataset was released for research purposes and only included scrambled information on patient identification, the study was exempted from informed consent from the subjects. Meanwhile, the study protocol has been approved by the Institutional Review Board of Fu-Jen Catholic University (FJU-IRB number: C104014).

### 2.2. Study Cohort

We assembled a study cohort comprising patients who received PPI prescriptions for the first time between 2000 and 2005 (*n* = 11,716) based on a new user design [25]. The first date of the prescription of PPIs was defined as the index date. Patients who met one of the following criteria were excluded: age younger than 40 or older than 80 years (*n* = 3455) and diagnoses of osteoporosis or hip fracture before the index date (*n* = 3398). In addition, patients with diagnoses of rheumatoid arthritis (*n* = 506), alcohol-related illness (*n* = 2585), chronic obstructive pulmonary disease (*n* = 3078), and use of glucocorticoid (*n* = 6166) during the study period were also excluded. Patients who underwent lansoprazole therapy were defined as the exposed cohort. Meanwhile, patients who received other PPI treatments, including omeprazole, pantoprazole, rabeprazole, and esomeprazole, were defined as the comparison cohort. To control for potential confounders between the two cohorts, we applied propensity score matching at a ratio of 1:4 for exposed and comparison cohorts. The propensity score was calculated for each patient by using a logistic regression model with covariates of age, sex, index date, and baseline comorbidities, including cerebral vascular disease (ICD-9-CM codes 430-438), chronic liver disease (ICD-9-CM codes 570-572), chronic kidney disease (ICD-9-CM code 585), hyperlipidemia (ICD-9-CM code 272), hypertension (ICD-9-CM codes 401-405), diabetes mellitus (ICD-9-CM code 250), malignant neoplasms (ICD-9-CM codes 140-208), and congestive heart failure (ICD-9-CM code 428). Finally, 655 patients with lansoprazole therapy and 2620 patients with other PPI treatments were included in the analysis. Follow-up was initiated from the index date until 31 December 2013, death (based on withdrawal from the NHI), or a first primary diagnosis of osteoporosis—whichever occurred first (Figure 1).

### 2.3. Identification of Exposure to PPIs

We identified patients who filled prescriptions for PPIs for the first time in the inpatient and ambulatory care order files between 1 January 2000 and 31 December 2005 as the study cohorts. We used Anatomic Therapeutic Chemical (ATC) codes to identify patients who were prescribed any PPI, including lansoprazole (ATC A02BC03), omeprazole (ATC A02BC01), pantoprazole (ATC A02BC02), rabeprazole (ATC A02BC04), and esomeprazole (ATC A02BC05).

### 2.4. Ascertainment of Osteoporosis

The outcome of interest was defined as a primary diagnosis of osteoporosis (ICD-9-CM codes 733.0 and 733.1). The criteria for diagnosing osteoporosis were based on the finding of a T-score ≤ −2.5 at the lumbar spine, femoral neck, or hip by a bone-mineral density test or clinical evidence of a low-trauma fracture in the vertebrae, proximal humerus, hip, or pelvis on a radiograph [26]. In order to identify patients with osteoporosis with sufficient accuracy, we identified patients as having the primary diagnosis of osteoporosis during at least two outpatient visits or one inpatient hospitalization.

### 2.5. Statistical Analysis

Chi-square and *t* tests were used to assess the differences in the distributions of categorical and continuous variables between the study cohorts. In addition, the Kaplan–Meier method was used to estimate the cumulative incidence of osteoporosis for study cohorts. Subsequently, the log-rank test was used to assess statistical differences in the cumulative incidence of osteoporosis between the cohorts. Furthermore, Cox proportional hazard regression models were performed to compute hazard ratios (HRs) with 95% confidence intervals (CIs) to evaluate the association between the clinical use of lansoprazole and the risk of osteoporosis. The proportional hazards assumption of the Cox models was evaluated with the log minus log plot of survival and Schoenfeld residuals method [27], which revealed no significant departures from proportionality in hazards over time. All of the data analyses were performed using SAS statistical software version 9.4 (SAS Institute, Cary, NC, USA). Statistical significance was set at 0.05, and all tests were 2-tailed.

## 3. Results

The baseline characteristics of the study cohorts are presented in Table 1. There were no significant differences in the distributions of age, sex, and comorbidities between the exposed and comparison cohorts due to the propensity score matching schemes.

In the follow-up period, the incidence rate of osteoporosis was 189.99 per 10,000 people in the lansoprazole-exposed cohort. Comparatively, the incidence rate was 307.98 per 10,000 people in the comparison cohort with exposure to other PPIs. The Kaplan–Meier curves for the cumulative incidence of osteoporosis between the two cohorts are shown in Figure 2. The cumulative incidence of osteoporosis was significantly higher in the comparison cohort with prescriptions of other PPIs than in the cohort with administration of lansoprazole (log-rank test, *p* < 0.001).

Compared with the comparison cohort with exposure to other PPIs, the lansoprazole-exposed cohort had a significantly reduced risk of osteoporosis (adjusted HR, 0.56; 95% CI, 0.46–0.68). Of note, the inverse association between the administration of lansoprazole and risk of osteoporosis was evident in both men (adjusted HR, 0.60; 95% CI, 0.44–0.82) and women (adjusted HR, 0.50; 95% CI, 0.38–0.66) and in various age groups [adjusted HR from 0.50 (95% CI, 0.36–0.71) to 0.66 (95% CI, 0.43–1.00)] (Table 2).

## 4. Discussion

In the current retrospective cohort study, our results showed that the administration of lansoprazole was associated with a significantly reduced risk of osteoporosis. Patients treated with lansoprazole had a 44% lower risk of osteoporosis than the comparison cohort who underwent other PPI therapy. Moreover, the inverse association between the clinical use of lansoprazole and the risk of osteoporosis was evident in both sexes and various age groups.

In 2005, a study found that gastric acid suppression by PPIs may result in a decreased calcium absorption and suggested that this might increase the risk of bone fracture [28]. Subsequently, multiple observational studies conducted in recent years have shed light on the relationship between PPIs and the risk of osteoporosis and osteoporotic fractures [9,10,11,12,13,29,30,31,32,33]. Early published studies in small numbers of subjects suggested that PPIs resulted in decreased calcium absorption after short exposures to the drug ranging from 3 days to 20 months [28,34,35]. Subsequently, numerous epidemiological studies reported that long-term PPI therapy was associated with an increased risk of osteoporotic fracture [9,10,11,12,13,29,30,31,32,33]. In addition, meta-analyses of observational studies demonstrated that the use of PPIs modestly increased the fracture risk [36,37]. Indeed, animal experiments showed that pantoprazole treatment for 12 weeks had a negative effect on bone metabolism in young male rats [38] and that pantoprazole could affect fracture healing in mice [39,40]. Previous studies have shown that both omeprazole and lansoprazole could induce arterial relaxation in a time-dependent manner, and this effect was associated with the regulation of intracellular calcium [41]. Schillinger and Sato demonstrated that pantoprazole could affect the uptake of Ca^2+^ in the sarcoplasmic reticulum by inhibiting SERCA, thus reducing the transient amplitude of calcium and myocardial contractility [42,43]. Aydin and Yurtsever hypothesized that omeprazole and lansoprazole may inhibit Rho-kinase, thus affecting Ca^2+^ regulation or blocking calcium channels to inhibit muscle contraction [44,45]. Thus, it seems that PPIs induce osteoporotic fracture via an effect on intracellular calcium homeostasis. By contrast, it has been noted that lansoprazole regulates adipocyte differentiation and decreases insulin resistance [17]. Previous studies reported an inverse relationship between insulin resistance and bone mineral density in a population-based study of young South Korean men [18], suggesting that insulin resistance was a negative predictor of bone health. Notably, one of the possible mechanisms of the induction of adipose differentiation by lansoprazole is via LXR [21]. Alterations in LXR have been observed to play a crucial role in physiological and pathological bone turnover [20]. LXRs orchestrate osteoblast/osteoclast crosstalk and contribute to pathologic bone loss. Thus, an LXR agonist would have therapeutic potential in treating bone diseases. It has been noted that lansoprazole acts as an LXR agonist [21]. Lansoprazole would increase bone density via the function of the LXR agonist. However, similar to other PPIs, the long-term use of lansoprazole may cause potential side effects, such as increased risks of recurrent C. difficile infection [46], microscopic colitis [47], vitamin B12 malabsorption [48], and kidney disease [49]. Taken together, although previous studies have indicated that other PPIs induce osteoporotic fractures, this study provides longitudinal evidence that the clinical use of PPI lansoprazole is associated with a reduced risk of osteoporosis.

The main strengths of this study include the use of a nationwide comprehensive prescription database rather than self-reported records, thereby minimizing recall bias. In addition, the NHIRD is a nationwide database of Taiwan’s general population, allowing for analyses to be performed in a real-life setting in an unselected patient population. However, some limitations of this study should be noted. Several important confounders, such as smoking, alcohol consumption, and nutritional status, were not available in the database. Therefore, it cannot be ruled out that there may be residual confounding in the current study. In addition, the use of a prescription database in this study did not permit an evaluation of compliance of PPI usage, as it was impossible to contact patients directly because of the anonymity of the records. Moreover, the actual bone mass density is not available and is not factored into the analysis.

## 5. Conclusions

This population-based cohort study demonstrated that lansoprazole was associated with a reduced risk of osteoporosis compared with other PPIs. The clinical implications of the present study require further investigations.

## Figures and Tables

**Figure 1 ijerph-19-15359-f001:**
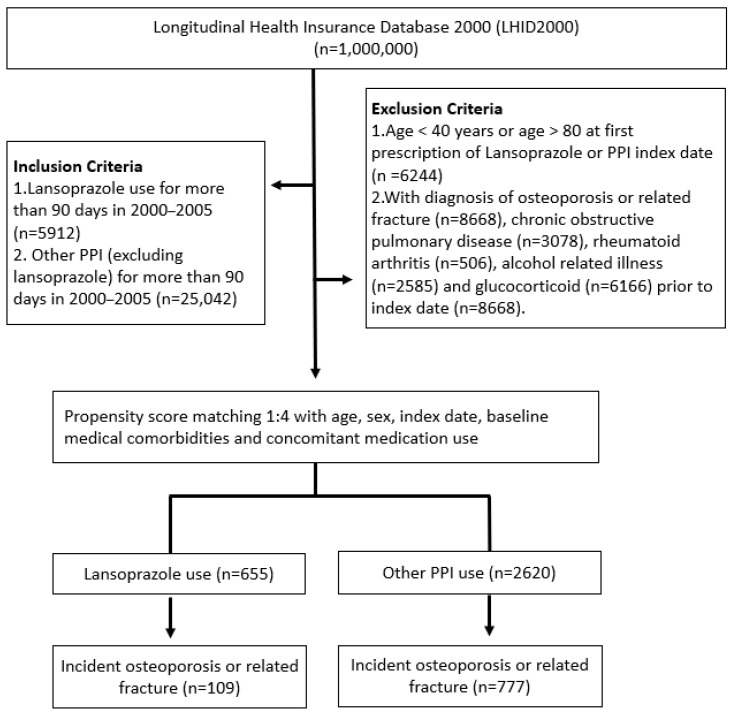
Flowchart of the study sample selection. PPI, Proton pump inhibitor.

**Figure 2 ijerph-19-15359-f002:**
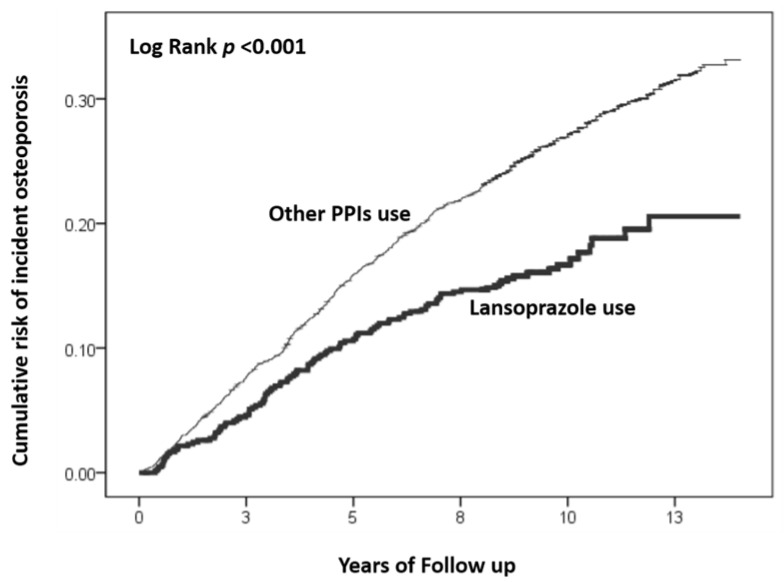
Kaplan–Meier curves for the cumulative risk of incident osteoporosis stratified by the use of lansoprazole and other PPIs with log-rank test.

**Table 1 ijerph-19-15359-t001:** Baseline characteristics of study cohorts.

	Study Cohorts		
Variable	Other PPIs	Lansoprazole	*p* Value
	(*n* = 2620)	(*n* = 655)	
Mean age	57.15 ± 11.32	57.73 ± 11.52	0.242
Sex (No., %)			1.000
Women	1020 (38.9%)	255 (38.9%)	
Men	1600 (61.1%)	400 (61.1%)	
Comorbidities (No., %)			
Cerebral vascular disease	456 (17.4%)	112 (17.1%)	0.854
Chronic liver disease	532 (20.3%)	149 (22.7%)	0.168
Chronic kidney disease	152 (5.8%)	38 (5.8%)	1.000
Hyperlipidemia	462 (17.6%)	99 (15.1%)	0.126
Hypertension	1266 (48.3%)	320 (48.9%)	0.807
Diabetes mellitus	631 (24.1%)	151 (23.1%)	0.580
Malignancy	359 (13.7%)	81 (12.4%)	0.370
Congestive heart failure	89 (3.4%)	27 (4.1%)	0.369

**Table 2 ijerph-19-15359-t002:** Association between administration of lansoprazole and other proton pump inhibitors (PPIs) and risk of osteoporosis.

Variable	No. of Subjects	No. of Osteoporosis Cases	Crude HR (95% CI)	Adjusted HR (95% CI)
Overall				
Other PPIs	2620	777	1.00	1.00
Lansoprazole	655	109	0.60 (0.49–0.74)	0.56 (0.46–0.68)
Sex				
Men				
Other PPIs	1600	323	1.00	1.00
Lansoprazole	400	47	0.66 (0.49–0.90)	0.60 (0.44–0.82)
Women				
Other PPIs	1020	454	1.00	1.00
Lansoprazole	255	62	0.54 (0.41–0.70)	0.50 (0.38–0.66)
Age (years)				
40–49				
Other PPIs	840	108	1.00	1.00
Lansoprazole	210	14	0.60 (0.34–1.05)	0.56 (0.32–0.98)
50–59				
Other PPIs	688	166	1.00	1.00
Lansoprazole	172	26	0.70 (0.47–1.07)	0.66 (0.43–1.00)
60–69				
Other PPIs	568	224	1.00	1.00
Lansoprazole	142	32	0.59 (0.41–0.86)	0.52 (0.36–0.76)
70–79				
Other PPIs	524	279	1.00	1.00
Lansoprazole	131	37	0.53 (0.37–0.74)	0.50 (0.36–0.71)

HR, hazard ratio; CI, confidence interval. Hazard ratios were adjusted for age, sex, index date, and comorbidities, including cerebral vascular disease, chronic liver disease, chronic kidney disease, hyperlipidemia, hypertension, diabetes mellitus, malignant neoplasms, and congestive heart failure.

## Data Availability

No additional data are available.

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
