# Peer review of "Clinical Use of Lansoprazole and the Risk of Osteoporosis: A Nationwide Cohort Study"

_ijerph, 2022, doi:10.3390/ijerph192215359_

Round 1

Reviewer 1 Report

line 92 page 2:"Patients who underwent lansoprazole treatments were defined as the exposed 92 cohort. Whereas, patients who received other PPI treatments, including lansoprazole, "¿? 

I think it is important to discuss more about the results, also about the negative ones found with lansoprazole.The discussion needs to highlight more the results found in a small number of studies in this regard.

Reviewer 2 Report

The authors determined the association between the use of lansoprazole and risk of osteoporosis. Though this study has some limitations that impede the applicability of the findings such as the potential confounding factors, the results of this study can be used as a basis for future research. Some issues should be addressed before being published.

Introduction

-        Please provide more details about specific PPI use and osteoporosis fractures particularly the mechanisms that the PPIs induce osteoporosis fractures.

-        Recent study has indicated that lansoprazole induced osteoporosis (ref: Cheng, Z., Liu, Y., Ma, M. et al. Lansoprazole-induced osteoporosis via the IP3R- and SOCE-mediated calcium signaling pathways. Mol Med 28, 21 (2022). This should also be described in the introduction

Methods

-        Several potential risk factors of osteoporosis were missing from the exclusion criteria such as smoking status, Rheumatoid arthritis, alcohol consumption, glucocorticoid use. Please provide reasons supporting the exclusion criteria. The authors only concern about the history of hip fractures.

Discussion

-        The discussion is not sufficient to distinguish the effect of lansoprazole on osteoporosis from other PPI. More mechanistic background of each PPI including lansoprazole should be discussed.

Reviewer 3 Report

The manuscript by Ming-Hsuan Chung et al claimed and concluded that proton pump inhibitor (PPI) lansoprazole treatment had a reduced osteoporosis risk as compared with those undergoing other PPIs treatments. However, the inclusion and exclusion criteria are simple and crude. In my opinion, the manuscript doesn’t meet the journal and the data of manuscript could not support the conclusion. I suggested rejected it.

Round 2

Reviewer 2 Report

The authors sufficiently responded to all the comments in the revised manuscript. I do not have further suggestions. 

Author Response

Thank you for your valuable comments.

Reviewer 3 Report

I still persist in my previous opinion.

Author Response

Thank you for your valuable comments.